# Differential Expression of Amaranth AtrDODA Gene Family Members in Betalain Synthesis and Functional Analysis of AtrDODA1-1 Promoter

**DOI:** 10.3390/plants14030454

**Published:** 2025-02-04

**Authors:** Huiying Xie, Jiajing Zeng, Wenli Feng, Wei Gao, Zhongxiong Lai, Shengcai Liu

**Affiliations:** 1Institute of Horticultural Biotechnology, Fujian Agriculture and Forestry University, Fuzhou 350002, China; xhy713061@163.com (H.X.); fwl9007@163.com (W.F.); gw1398982958@163.com (W.G.); laizx01@163.com (Z.L.); 2College of Horticulture, South China Agricultural University, Guangzhou 510642, China; ehe360722@163.com; 3Key Laboratory of Ministry of Education for Genetics, Breeding and Multiple Utilization of Crops, Fujian Agriculture and Forestry University, Fuzhou 350002, China

**Keywords:** *Amaranthus tricolor* L, betalain, *DODA* gene family, expression analysis, promoter activity

## Abstract

Betalains differ from anthocyanins, and they cannot coexist in the same plant under natural conditions. The L-DOPA 4,5-dioxygenase encoded by the *DODA* gene is a key step in the pathway of betalain biosynthesis in Caryophyllales plants. Amaranth is an important resource for the study and extraction of betalains. In order to clarify the function of *AtrDODA* family genes in betalain biosynthesis, we screened out three *AtrDODA* family gene members associated with betalains based on a genome database and RNA-seq databases of *Amaranthus tricolor*. Their characterization and expression pattern were further analyzed. The result of subcellular localization showed that all three AtrDODA members were located in the nucleus. Betacyanin and betaxanthin were promoted by paclobutrazol treatment in the leaves and stems of ‘Suxian No.1’ (red), while they were inhibited by gibberellin and darkness, which were consistent with the gene expression pattern of *AtrDODAs*. After heterologous transformation of the *AtrDODA1-1* promoter into tobacco with GUS staining analysis, the promoter activity of *AtrDODA1-1* of ‘Suxian No.1’ (red) amaranth was significantly higher than that of ‘Suxian No.2’ (green) amaranth. Furthermore, we analyzed the promoter activity of *AtrDODA1-1* by GUS staining and qRT-PCR after sprayed exogenous MeJA and GA_3_ on the *AtrDODA1-1* promoter transformed tobacco plants. The results showed that *AtrDODA1-1* responded to plant hormones. This study could lay a foundation for revealing the biological functions of the amaranth *DODA* gene family, and provide new clues for the molecular mechanism of betalain synthesis.

## 1. Introduction

Natural plant pigments are categorized into four types: chlorophyll, carotenoids, anthocyanins, and betalains. Under natural conditions, anthocyanins and betalains cannot coexist in the same plant. Betalains, as significant characteristic pigments, are currently found only in some plants of the Caryophyllales order and the fungus *Amanita muscaria*, such as *Portulaca oleracea* [1], *Mirabilis jalapa* [2], *Bougainvillea* [3], *Suaeda salsa* [4], *Beta vulgaris* [5], *Hylocereus polyrhizus* [6], and *Amaranthus tricolor* [7,8].

The biosynthetic pathway of betalains includes the formation of betacyanins and betaxanthins. They originate from tyrosine to stable betalains, involved in at least four enzymatic reaction steps [9]. The key enzymes include tyrosinase (TYR/PPO), cytochrome P450 enzymes (CYP450), dopa 4,5-dioxygenase (DODA), and glucosyltransferase (GT) [6]. Tyrosine is the precursor for betalain synthesis, which is converted into dopamine (L-DOPA) by the catalytic action of tyrosinase; subsequently, dopamine is catalyzed to dopamine quinone, and then spontaneously forms cyclo-DOPA [2,10]. L-DOPA can be directly converted to cyclo-DOPAby cytochrome P450 (CytP450), according to recent research [11]. Meanwhile, L-DOPA can also be converted into open cyclo-DOPA by 4,5-dopa dioxygenase (DOD), which then spontaneously forms betalamic acid. Betalamic acid is the core skeleton for betalain synthesis and can spontaneously combine with amino acids or amines to form betaxanthins; it can also combine with the imide ion of cyclo-DOPA to form betadin, and then forms betacyanins via glucosyltransferases (GTs) [12]. Compared to the synthesis mechanisms of carotenoid and anthocyanin, the biosynthesis mechanism of betalains is yet unclear [13]. In order to further explore the mechanism of betalain biosynthesis, it is necessary to study the key enzymes involved in betalain biosynthesis and their transcriptional regulation.

In plants, *DODA* belongs to the *LigB* gene family, which can be divided into two types, Class I and Class II. Class I contains *DOD-like* genes, while Class II contains the *DODA* gene with DODA enzyme activity and function [14]. The first *DOD* gene was cloned from *Amanita muscaria* [15]. DODA, the first characterized member of a novel family of plant dioxygenases, was cloned from Portulaca grandiflora by Chrisfinet et al. [16]. With the release of the beet genome [17], the important role of the *DODA* gene in betalain synthesis has been elucidated [5]. The *SsDODA* promoter has been cloned from *Salicornia europaea*, which contains a low-temperature response element (LTR) (CCGAAA) [4]. *SsDODA* gene expression and betacyanin synthesis are influenced by low temperature [18]. The content of betalain in *Bougainvillea* was found to be consistent with *DODA* expression levels [19]. Hatlestad et al. found that the expression level of *BvDODA1* is lower in white beets than in red beets [5], and silencing *BvMYB1* downregulates the expression of the *BvDODA1* gene and reduces the accumulation of betacyanin pigments [20]. Expression of the DODA of *Amanita* and *Portulaca grandiflora* could induce the production of betalains in the petals of *Antirrhinum majus* and *Solanum tuberosum* [15]. Overexpression of the *DOD-like* gene from *Portulaca grandiflora* could promote the accumulation of betalains in the petals of white *Portulaca* [16]. The overexpression of the *CYP76AD* gene from mushroom and the *DOD* gene from *Mirabilis jalapa* in tobacco and *Arabidopsis* successfully induced the production of betalains [21].

Betalain synthesis is influenced not only by intrinsic genetic factors such as genes, transcription factors, or promoters, but also by various environmental factors, including light (light quality, intensity, and photoperiod) [22,23], hormones (auxins, gibberellins, cytokinins, and abscisic acid, etc.) [24,25,26,27,28], inorganic salts [29,30,31,32,33], oxidative stress [34], mechanical damage [35], and temperature [36,37,38], all of which affect betalain synthesis. Light is an important factor in regulating betacyanin synthesis. Blue light increases the content of betaxanthins, while white or red light can promote the accumulation of betacyanins and betaxanthins in the seedlings of *Amaranthus tricolor* and cockscomb [39]. Exogenous GA_3_ (gibberellin) regulates the synthesis of betacyanins by inhibiting phytochrome [40]. GA_3_ inhibits the synthesis of betacyanins in *Amaranthus caudatus*. However, the betalain precursors L-tyrosine and L-dopa applied on gibberellin-treated seedlings completely recovered betacyanin synthesis, while gibberellin-induced growth enhancement remained unaltered. Using 50 and 100 mM NaCl could significantly improve plant growth and increased the content of betalains in *Mesembryanthemum crystallinum*, but 150–300 mM NaCl showed the opposite effect [41]. Salt concentrations of 2% and 3% could promote the accumulation of betalains [42]. Additionally, 20 g/L NaCl significantly increased the content of betalains in *Suaeda salsa* [43].

*Amaranthus tricolor* L., commonly known as amaranth, is an annual dicotyledonous herbaceous plant belonging to the Amaranthaceae family within the Caryophyllales order. It is rich in betalains in its stems and leaves, and is considered a highly nutritious functional vegetable. Amaranth is primarily distributed in Central and South America, as well as in some regions of Asia and Africa [44]. *Amaranthus* is an important resource for the extraction and study of betalains because of its rapid growth, high biomass, and rich pigment content. Betalain has become a research hotspot. However, the role of *AtrDODA* in the biosynthesis of betalains in amaranth is unknown. Furthermore, the differences in accumulation mechanisms of betalains between red and green amaranth are not yet clear. Therefore, we screened out the *DODA* family gene members associated with betalains based on a genome database and RNA-seq databases of *Amaranthus tricolor*, and further analyzed their characterization and expression pattern. Meanwhile, the promoter activity difference of *AtrDODA1-1* was analyzed between ‘Suxian No.1’ (red) amaranth and ‘Suxian No.2’ (green) amaranth. Furthermore, we analyzed the promoter activity of *AtrDODA1-1* after sprayed exogenous MeJA and GA_3_ on the *AtrDODA1-1* promoter transformed tobacco plants. This study lays a foundation for revealing the biological functions of the amaranth *DODA* gene family, and provides new clues for the molecular mechanism of betalain synthesis.

## 2. Results

### 2.1. Identification of AtrDODA Gene Family Members and Analysis of Physicochemical Properties

Based on the amaranth genome and bioinformatics analysis, three complete and conserved domain-containing *AtrDODA* family members were ultimately identified in amaranth. Referring to the annotation and nomenclature of *DODA* family members in Caryophyllales [14], the three members were named *AtrDODA1-1*, *AtrDODA1-2*, and *AtrDODA2-1*. The protein sequence lengths were 286 amino acids (aa), 268 aa, and 269 aa, respectively. Isoelectric point analysis revealed that AtrDODA1-1 (pI = 6.03) and AtrDODA1-2 (pI = 6.21) were acidic proteins, but AtrDODA2-1 (pI = 7.26) was a basic protein. The instability coefficient revealed that AtrDODA1-2 was a stable protein, while AtrDODA1-1 and AtrDODA2-1 might be unstable proteins. Subcellular localization prediction showed that AtrDODA1-1 was localized in the nucleus, while AtrDODA2-1 and AtrDODA1-2 were localized in both the nucleus and the cytoplasm. The detailed physicochemical properties are listed in Appendix A.

### 2.2. Analysis of the Structure, Chromosomal Localization, and Motif Composition of AtrDODA

Analysis of the gene structure of the *AtrDODA* family revealed that all three family members contained one 4,5 DOPA dioxygenase domain (Figure 1A), and each member consists of three exons and two introns (Figure 1B), indicating that the gene structure of the three members is relatively consistent.

Genome chromosomal location analyses indicated *AtrDODA1-1* was located on chromosome 16, while *AtrDODA2-1* and *AtrDODA1-2* are located on the same segment of chromosome 12 (Figure 1C), suggesting that they might be chromosomal tandem duplication or segmental duplication.

The conserved motif analysis showed that all three AtrDODA family members’ amino acid sequences contained motif 1 to 8 with the same order (Figure 1D), indicating that these 8 motifs are relatively conserved in AtrDODA proteins. Additionally, motif 9 was found in AtrDODA1-1 and AtrDODA1-2, except for AtrDODA2-1. Furthermore, motif 10 only existed in AtrDODA1-1, which might result in the member functioning differently from AtrDODA2-1 and AtrDODA1-2.

### 2.3. Collinearity Analysis and Construction of a Phylogenetic Tree

We conducted a collinearity analysis to understand the evolution relationships of amaranth *DODA* family members with other plants of the order Caryophyllales, including *Beta vulgaris*, *Hylocereus undatus, Portulaca oleracea,* and *Bougainvillea spp*. As shown in Figure 2A, amaranth had two genes (*AtrDODA2-1* and *AtrDODA1-1*) on two chromosomes (Chr 12 and Chr 16) forming two pairs of collinearity with two genes (transcript:KMT13936 and transcript:KMT18891) on two chromosomes (Chr 2 and Chr 4) of *Beta vulgaris*. However, there was no collinearity between amaranth and *Hylocereus undatus*. The results suggested that the genetic relationship was closer between amaranth and beet than between amaranth and *Hylocereus undatus*. In addition, the collinearity of *DODA* among amaranth, purslane, and *Rhododendron leucoides* was analyzed (Figure 2B). The result showed *AtrDODA2-1* on Chr12 formed three pairs of collinearity with three genes (evm.model.LG06.1319, evm.model. LG18.1890 and transcript:KMT18891) on three chromosomes (Chr6, Chr18, and Chr20) of *Portulaca oleracea*. Meanwhile, *AtrDODA2-1* on Chr 12 formed two pairs of collinearity with two genes (Bou_130730 and Bou_74018) on two chromosomes (Chr17 and Chr29) of *Rhododendron leucocephala*. This indicates that the genetic relationship was closer between amaranth and purslane than between amaranth and *Rhododendron leucocephala*. In addition, *AtrDODA1-1* existed collinearity only in the amaranth and beet, implying *DODA1-1* had a similar function between amaranth and beet.

The values of Ka (non-synonymous mutation rate) and Ks (synonymous mutation rate) were calculated from different species (Appendix A). The Ka/Ks ratio is commonly used to determine whether a gene encoding a protein is subject to selective pressure. A Ka/Ks ratio  >  1 generally indicates a positive selection effect, while a Ka/Ks ratio  =  1 suggests neutral selection, and a Ka/Ks ratio  <  1 implies purifying selection. In this study, the Ka/Ks values of the beet (transcript: KMT13936 and transcript: KMT18891) homologous gene pairs with *AtrDODA1-1* and *AtrDODA2-1* from *Amaranthus tricolor* were less than 1. Similarly, the Ka/Ks ratios of the *DODA* homologous gene pairs between *Amaranthus tricolor* and *Portulaca oleracea*, as well as between *Amaranthus tricolor* and *Bougainvillea*, were also less than 1. These results indicate that the *DODA* pairs from different species evolved under the pressure of purifying selection. Additionally, the Ka/Ks ratio of *AtrDODA2-1* and *AtrDODA1-2*, which were located on chromosome 12 of *Amaranthus tricolor*, was also found to be less than 1, further indicating that the *AtrDODA* gene is under purifying selection.

A total of 70 members of *DODA* of different plants can be classified into four different clustering groups (Figure 2C). *AtrDOA1-1* was closely related to *Beta vulgaris DODA1*, *AtrDOA2-1* is closely related to *Amaranthus hypochondriacus LigB*, and *AtrDODA1-2* is closely related to *Amaranthus tricolor DOD* and *Amaranthus hypochondriacus DODA*. *AtrDOA1-1* is in the same group as *AtrDOA1-2*, implying they might perform similar functions.

### 2.4. Multiple Comparison Analysis with Protein Sequences

Analysis of multiple sequence comparisons of DODA proteins revealed that the LigB homologues could be classified into two categories according to the sequence alignment of catalytic domains and neighbouring regions (Figure 3). AtrDODA1-1 and AtrDODA1-2 are more similar to the DOD gene and its homologous genes in Class II, which contain important acidic residues (HPLDETP) for DOD activity/DOPA recognition, whereas AtrDODA2-1 is similar to the LigB family homologous gene in Class I.

### 2.5. Analysis of Cis-Acting Elements of the AtrDODA Promoter

In order to understand the response of *AtrDODAs* to environmental stress and the hormone response, cis-regulatory elements were predicted using PlantCARE (Figure 4). Among the nine predicted response elements, all family members were involved in photoresponsive elements and ethylene responsive elements. However, there are some differences in the response element number of *AtrDODAs* members. Only the promoters *red-AtrDODA1-1p*, *red-AtrDODA2-1p*, and *green-AtrDODA2-1p* contained MYB binding sites involved in light response elements. The promoters *red-AtrDODA1-1p*, *green-AtrDODA1-1p*, *red-AtrDODA2-1p*, and *green-AtrDODA2-1p* contained MYB response elements, suggesting that *AtrDODA1-1p* and *AtrDODA2-1p* might be regulated by MYB binding factors. The promoters *red-AtrDODA2-1p*, *green-AtrDODA2-1p*, and *green-AtrDODA1-2p* contained abscisic acid response elements. In addition, salicylic acid response elements only presented in *red-AtrDODA2-1p* and *green-AtrDODA2-1p*; gibberellin response elements were only found in *red-AtrDODA1-1p* and *green-AtrDODA1-1p*; and the promoters *red-AtrDODA1-1p*, *green-AtrDODA1-1p*, *red-AtrDODA2-1p*, and *green-AtrDODA2-1p* contained jasmonic acid response elements. The above analysis indicated that the promoter of different *DODA* gene family members or the same gene in two different amaranth varieties contained different types and quantities of response elements, suggesting different functions in responding to external conditions or hormonal signals. In addition, different *cis*-acting elements are also classified based on stress response, hormone response, growth response, light response, and others (Appendix A).

### 2.6. Expression Pattern of AtrDODA Based on TPM Values

Based on the transcriptome database TPM values of ‘Suxian No.1’ amaranth treated with gibberellin and paclobutrazol, and then cultured under blue light, white light, and dark conditions (Figure 5A), we analyzed the expression patterns of the *AtrDODA* family genes. *AtrDODA2-1* was significantly upregulated under dark-gibberellin treatment and showed low expression under other treatments. *AtrDODA1-1* was significantly downregulated under dark-gibberellin treatment and significantly upregulated under white light–paclobutrazol treatment. *AtrDODA1-2* was significantly upregulated under white light–gibberellin and white light–paclobutrazol treatments, and significantly downregulated under dark-paclobutrazol treatment. According to the transcriptome database TPM analysis of the red sections and green sections of the same leaf in ‘Dahong’ amaranth, *AtrDODA1-1* and *AtrDODA1-2* are significantly upregulated in the red sections of the ‘Dahong ‘ amaranth leaf, while *AtrDODA2-1* is significantly upregulated in the green sections of the ‘Dahong ‘ amaranth leaf (Figure 5B).

### 2.7. AtrDODA Gene qRT-PCR Analysis

To further clarify the expression patterns of the *AtrDODA* family genes, qRT-PCR was used to detect the relative expression levels of the three *AtrDODA* gene members. The results showed that there were differences in plant phenotypes under different treatments (Figure 6A), and the expression patterns of the three *DODA* genes also were different in the different tissues of ‘Suxian No.1’ and ‘Suxian No.2’. *AtrDODA1-1* and *AtrDODA1-2* showed high expression in the leaves and roots of ‘Suxian No.1’ (red) amaranth, respectively. *AtrDODA2-1* showed high expression in the leaves of ‘Suxian No.2’ (green) amaranth (Figure 6B–D).

Furthermore, we investigated the expression patterns of the three *DODA* genes in the stems and leaves of ‘Suxian No.1’under different light qualities, hormones, and salt treatments. We found that there were some differences in the expression patterns of the three members in leaves and stems under different treatments. *AtrDODA1-1* and *AtrDODA2-1* showed high expression in leaves and stems under paclobutrazol treatment; *AtrDODA1-2* showed high expression in stems under paclobutrazol treatment. Under salt treatment, *AtrDODA1-1* showed high expression in the control (0 mM) and 50 mM NaCl, while *AtrDODA2-1* and *AtrDODA1-2* showed higher expression in the control (0 mM).

### 2.8. Betalain Content

To further elucidate the relationship between betalain content and gene expression, the content of betacyanin and betaxanthin in all samples was measured. The results showed that the plant phenotypes corresponded to the color of the pigments in the centrifuge tubes (Figure 7A–F). The content of betacyanin and betaxanthin in the leaves of ‘Suxian No.1’ (red) amaranth was 10.13 times and 9.11 times of that in the root, respectively. There was no difference in the content of betaxanthin in different parts of ‘Suxian No.2’ (green) amaranth, but the content of betaxanthin in leaves was 3.61 times of that in roots and stems (Figure 7G,H). The contents of betacyanin and betaxanthin in leaves and stems of ‘Suxian No.1’ were increased significantly under the paclobutrazol treatment, and their contents were 1.75 times, 6.00 times, 1.70 times, and 4.00 times of that under white light treatment, respectively. Under the treatment of gibberellin, the content of betacyanin in the leaves and stems of ‘Suxian No.1’ was significantly lower than that without gibberellin, and the content of betaxanthin in the leaves was also significantly lower than that without gibberellin, but there was no difference in the content of betaxanthin in the stems. The contents of betacyanin and betaxanthin in stems under dark treatment were lower than those under white light treatment.

### 2.9. Subcellular Localization of the AtrDODA Proteins

Using PCR technology, the recombinant vectors pRI101-AN-*35S*::AtrDODA1-1-eGFP, pRI101-AN-*35S*::AtrDODA2-1-eGFP, and pRI101-AN-*35S*::AtrDODA1-2-eGFP were successfully constructed (Appendix A). Subcellular localization results showed green fluorescence of containing the target fragments AtrDODAs was detected exclusively in the nuclei (as evaluated by 4′,6-diamidino-2-phenylindole (DAPI) staining) (Figure 8), and this green fluorescence co-localized with the blue fluorescence of the nuclear localization DAPI signal, while onion epidermal cells transformed with the empty vector exhibited green fluorescence around the cytoplasm, cell membrane, and nuclei. These results indicate that AtrDODA1-1, AtrDODA2-1, and AtrDODA1-2 were all localized in the nucleus.

### 2.10. Promoter Activity of AtrDODA1-1 in ‘Suxian No.1’ and ‘Suxian No.2’ Amaranth

The promoter sequences of *AtrDODA1-1* from ‘Suxian No.1’ and ‘Suxian No.2’ amaranth were cloned separately and named *Red-AtrDODA1-1pro* and *Green-AtrDODA1-1pro*, respectively. Sequence alignment analysis revealed the two promoter sequences were different, and the similarity between them was 87.50% (Appendix A). In order to analyze whether there are differences in promoter activity between *Red-AtrDODA1-1pro* and *Green-AtrDODA1-1pro*, we constructed the pCAMBIA1301-*AtrDODA1-1pro*::GUS recombinant vector (Appendix A). After infecting tobacco leaves with Agrobacterium, GUS staining and qRT-PCR analysis were performed. The GUS sites in the transgenic plant leaves showed different degrees of blue color. The GUS staining intensity in the leaves of *pRed-AtrDODA1-1pro*::GUS transgenic plants was similar to that of the *p35S*::GUS plants, while the GUS staining intensity in the leaves of *pGreen*-*AtrDODA1-1pro*::GUS transgenic plants was the lightest (Figure 9). Meanwhile, GUS enzyme activity was consistent with GUS staining results. In conclusion, the activity of *Red-AtrDODA1-1pro* of ‘Suxian No.1’ was higher than that of *Green-AtrDODA1-1pro* of ‘Suxian No.2’.

### 2.11. Response of AtrDODA1-1 Promoters from ‘Suxian No.1’ and ‘Suxian No.2’ Amaranth to Exogenous Hormones Treatment

By analyzing the potential cis-regulatory elements of the *AtrDODA1-1* promoter in ‘Suxian No.1’ and ‘Suxian No.2’, we found that both of them contained MeJA and GA_3_ response elements (Figure 10A). GUS staining showed that the GUS blue spots of *pRed-AtrDODA1-1 pro*::GUS almost covered the entire leaf surface under 200 μM MeJA treatment for 24 h, whereas the GUS blue spots gradually faded over time under GA_3_ treatment (Figure 10B). The GUS blue spots of *pGreen-AtrDODA1-1 pro*::GUS reached the peak under 100 μm MeJA treatment for 12 h and 100 μm GA_3_ treatment for 12 h (Figure 10C). Subsequently, the GUS enzyme activity of the leaves at the corresponding stage was evaluated with Image J software. The results showed that GUS enzyme activity was consistent with the distribution of GUS blue spots. Furthermore, the expression of GUS gene in different leaves at different time points after hormone treatment was analyzed by qRT-PCR. The results showed that the *Red-AtrDODA1-1pro* GUS reporter gene showed significantly high expression at 200 μm-24 h after MeJA treatment, and the *Green-AtrDODA1-1pro* GUS reporter gene showed significantly high expression at 100 μM-12 h and 24 h (Figure 10D). After GA_3_ treatment, the expression of the GUS gene under the control of *Red-AtrDODA1-1pro* showed a downtrend over time, while *Green-AtrDODA1-1pro* exhibited significant high expression of the GUS gene at 100 μM-12 h and lower expression at other stages. Overall, under the treatment of MeJA and GA_3_, the expression levels of the GUS gene and GUS enzyme activity were consistent with the results of GUS staining, indicating that *AtrDODA1-1* responds to MeJA and GA_3._

## 3. Discussion

### 3.1. Bioinformatics Analysis of Potential Functions of the AtrDODA Gene Family

It is known that betalain synthesis in plants originates from tyrosine, which then is then converted into betalain under the action of tyrosinase (TYR), CYP76AD, 4,5 dopa dioxygenase (DODA), and glycosyltransferase (GT) [2]. Genome sequencing technology is currently an effective means to obtain species genome information. In this study, a total of *DODA* genes were identified using the amaranth genome, which was similar in number to that of beet (2) [45], and pitaya (3) [46], but less than that of quinoa (18) [47]. This might be due to the conserved evolution of the *DODA* family in amaranth, beet, and pitaya. During the evolution of the quinoa *DODA* family, gene segment duplication and tandem duplication occurred.

Analysis of the gene structure of the three *AtrDODA* members revealed that they all contained three exons and two introns, which was similar to the *DODA* gene structure in beet and dragon fruit. However, the number of introns in quinoa *DODA* genes ranges from 2 to 15, and the number of exons ranges from 2 to 16. Most quinoa *DODA* members contain one to three exons [47]. In addition, *DODA* in *Amanita muscaria* contains five short introns, which might be due to evolutionary differences between species [48]. By comparing the sequences of the catalytic domain of LigB homologues and their adjacent regions, it was found that the amino acid residue of AtrDODA1-1 and AtrDODA1-2 in Class I was HPLDETP and HPSNNTP, respectively, and the amino acid residue of AtrDODA2-1 in Class Ⅱ was HNLRSLD, which is consistent with previous studies [14,49]. In phylogenetic tree construction, a bootstrap value of 90% or above indicates strong support for the specific grouping at that node, suggesting that the relationships among the taxa within that group are well supported by the analyzed data. In contrast, bootstrap values between 50% and 90% indicate moderate support, while values below 50% imply relatively low confidence in the grouping. Analysis of the phylogenetic tree showed that *AtrDODA1-1* and *AtrDOA1-2* were clustered in subgroup II, but *AtrDODA1-1* and *BvDODA1* was clustered in the same branch. *AtrDODA1-2* was clustered in the same branch with *Amaranthus tricolor DOD* and *Amaranthus hypochondriacus DODA*; the *AtrDOA2-1* in subgroup IV was clustered in the same branch with the *Amaranthus hypochondriacus LigB*. These bootstrap values were all 99, indicating that the analyzed data were credible. These results suggested that *DODA* family members within the same class might perform similar functions.

In addition, the protein tertiary structure model showed that the protein similarity reached 67.04% between *Amaranthus tricolor DODA1-1* and *Beta vulgaris DODA1*. The similarity reached 50.96% and 95.54% between *Amaranthus tricolor DODA1-2* and *Beta vulgaris DODA1*, and between *Amaranthus tricolor DODA2-1* and *Amaranthus hypochondriacus DODA2* (Appendix A). The protein tertiary structure model was consistent with the clustering members of the phylogenetic tree, which indicates that the similar proteins in different species have similar functions.

### 3.2. Correlation Between AtrDODAs Gene Expression and Betalain Content

Previous studies have reported that the transient overexpression of the *Hylocereus pitaya DODA1* gene in tobacco increased the content of betalains, while the silencing of the *DODA1* gene in pitaya scales reduced the betalain content. This accumulation phenotype is consistent with the phenotypes observed in previous studies where both homologous and heterologous expression of *DODA1* could lead to the formation of betalains [5,15]. However, overexpression *HpDODA2* and supplementation with L-DOPA substrate in *Nicotiana benthamiana* and *Arabidopsis thaliana* did not result in any altered pigment phenotype [50]. Therefore, the researchers infer that the 4, 5-dopa dioxygenase activity influenced betalain synthesis [51]. *HpDODA2* may not have or easily achieved the enzymatic activity for the synthesis of betalains [50]. *BvDODA1* in beet plays an important role in the betalain biosynthetic pathway, while BvDODA2 cannot promote betalain synthesis. However, targeted mutagenesis of *BvDODA2* could promote betalain synthesis [12]. In this study, we determined the contents of betacyanin and betaxanthin in the roots, stems, and leaves of ‘Suxian No. 1’ (red) and ‘Suxian No. 2’ (green), and found that betacyanin and betaxanthin were significantly accumulated in the leaves of ‘Suxian No.1’ (red). Betaxanthin was significantly accumulated in the leaves of ‘Suxian No. 2’ (green). The results indicated that there are differences in the betalain content between different varieties and different plant tissues, and an increase in betacyanin content may also be accompanied by an accumulation of betaxanthin content. Additionally, *AtrDODA1-1* and *AtrDODA1-2* were highly expressed in the leaves and the roots of ‘Suxian No. 1’ (red). *AtrDODA2-1* was highly expressed in the leaves of ‘Suxian No. 2’ (green). The results showed that the expression patterns of the three members were different in different tissues of the amaranth plant. Furthermore, we found that there was a positive correlation between the expression of *AtrDODA1-1* and the content of betalain, and the *AtrDODA1-1* gene was highly expressed in the tissues with high content of betalain, suggesting that it might have a similar biological function of *BvDODA1* and *HpDODA1*.

It is noteworthy that the homologous gene *AtrDODA1-2* was highly expressed in the root of ‘Suxian No. 1’ (red), but betalain was not accumulated in the root. Further studies are needed to determine whether there is a new mechanism of betalain synthesis by regulating the *AtrDODA1-2* gene, or if *AtrDODA1-2* has more functions. We found that the expression level of *AtrDODA2-1* in the leaves of ‘Suxian No. 2’ (green) was significantly positively correlated with the content of betaxanthin, suggesting that *AtrDODA2-1* played an important role in the synthesis of betaxanthin in the leaves of ‘Suxian No. 2’ (green). Studies have shown that *Amaranthus hypochondriacus DODA2* in has a closer phylogenetic relationship with the *DODA-like* gene in anthocyanin synthesis in plants [52], and *AtrDODA2-1* and *Amaranthus hypochondriacus DODA2* might have similar functions because of the high protein similarity.

Environmental factors such as light quality, plant hormones, and stress can affect the synthesis of betalains. Light is one of the important environmental factors influencing betalain synthesis. Different light qualities and intensities can impact the accumulation of betalains in plants. Red light could significantly increase the content of betacyanin and betaxanthin [53,54]. The content of betalain in callus of Alternanthera brasiliana was promoted under blue and white light condition, while red light had no effect [55]. The red beet seedlings were irradiated with red, green, and blue light separately, indicating that blue light was beneficial for the accumulation of betalain, while red light and green light inhibited betalain accumulation [56]. In this study, we found that under blue light treatment, the content of betacyanin and betaxanthin in the leaves and stems of ‘Suxian No.1’ (red) amaranth was not significantly different from that under white light treatment, while dark treatment significantly inhibited the accumulation of betacyanin and betaxanthin. This is inconsistent with the results of light quality affecting the content of betacyanin in the cotyledons of *Suaeda salsa* seedlings [57], indicating that the regulation of light quality on the synthesis of betalain is complicated. Quantitative expression analysis also revealed differences in the expression of *AtrDODA* under different light quality treatments. *AtrDODA1-1* expression was significantly lower in the dark than under white and blue light, with no significant difference in expression between white and blue light treatments. The expression trends of *AtrDODA1-2* and *AtrDODA2-1* were consistent, and the expression in leaves under blue light was significantly higher than that under other treatments, and the expression in leaves in the dark was not significantly different from that under other treatments. It was reported that *Suaeda salsa DODA* had different expression levels in response to different light properties, and light treatment (white light, red light, and blue light) significantly increased *DODA* expression compared to dark. The response of *DODA* to different light qualities was similar to that of betalain accumulation, indicating that the mRNA expression level of *DODA* affected by different light quality was consistent with the increase of betalain accumulation [58]. This result was similar to that of the expression pattern of *AtrDODA1-1* in amaranth, but different from that of *AtrDODA1-2* and *AtrDODA2-1*, suggesting that the expression patterns of the three *AtrDODA* members in amaranth were different under different light quality.

Gibberellin can regulate the growth and development of plants and the synthesis of secondary metabolites. Exogenous GA_3_ significantly reduce the level of anthocyanins induced by low temperatures, while PAC increases anthocyanin levels, suggesting that GAs can negatively regulate low-temperature-induced anthocyanin accumulation [59]. Exogenous GA could control the synthesis of betacyanins by inhibiting phytochromes [40]. GA_3_ inhibits the synthesis of betacyanins in amaranth, but when GA_3_-treated seedlings are sprayed with the pigment precursors L-tyrosine and L-dopa, the synthesis of betacyanins suppressed by GA_3_ is completely recovered [60]. The content of betalains in amaranth decreases with increasing concentrations of GA_3_, yet the expression levels of DODA genes exhibit an opposite trend, increasing with the addition of GA_3_ [61]. We treated ‘Suxian No.1’ amaranth with gibberellin and paclobutrazol. The results showed that paclobutrazol significantly promoted the accumulation of betacyanin and betaxanthin in stems and leaves, while the gibberellin treatment showed the opposite trend, which was consistent with previous studies [60,61,62]. Furthermore, we analyzed the correlation between the expression of *DODA* gene family members and betalain content under the condition of combined light quality and hormones. The expression of the *AtrDODA1-1* and *AtrDODA2-1* genes was positively correlated with the content of betalain, both of which were highly expressed under the induction of paclobutrazol, which resulted in high levels of betalains. In conclusion, the mRNA levels of *AtrDODA1-1* and *AtrDODA2-1* were consistent with the tendency of light quality and exogenous hormones to regulate the content of betacyanin content, suggesting that these factors regulated the expression of *AtrDODA1-1* and *AtrDODA2-1,* thereby controlling the synthesis of betalain in amaranth.

### 3.3. The Different Activity of the DODA1-1 Promoter in Red and Green Amaranth May Affect Betalain Synthesis

In general, the regulatory region of eukaryotic genes contains promoter regions and various regulatory elements. A promoter is a DNA sequence located upstream of a gene that recognizes and binds RNA polymerase and general transcription factors. Natural variations in cis-regulatory regions often affect plant phenotypes by altering gene expression [63,64,65,66]. The W-box sequence, a key cis-acting element in the promoter of *WRKY33* gene, was found to be TTGACT in wild cold-resistant tomato and TTGATT in cultivated tomato. The single base variation of this promoter resulted in the inability of *WRKY33* to achieve self-transcriptional regulation and protein accumulation under low-temperature stress in cultivated tomato, thus failing to activate downstream signaling pathways related to cold tolerance, and ultimately resulting in cold sensitivity in cultivated tomato. This result indicated that nucleotide polymorphisms in cis-regulatory elements were closely related to differences in abiotic stress resistance during plant evolution [64]. Moreover, single base variation in the promoter region of different varieties may also be closely related to biotic stress [63,66]. By analyzing the promoter of MdhpRNA277 in apple leaf spot resistance and susceptible varieties, a single base mutation (G-T) was discovered at the -1186 bp in motif b. The transcription factor MdWHy in disease-resistant varieties could not bind to the motif b of the MdhpRNA277 promoter, which prevented the normal transcription of MdhpRNA277. As a result, mdm-siR277-1 and mdm-siR277-2 were not induced, and the expression levels of MdRNL1/2/3/4/5 were high, thus exhibiting resistance to apple leaf spot disease [63].

We found that there were differences, with a similarity of 87.50%, in the *AtrDODA1-1* promoter sequence between ‘Suxian No.1’ and ‘Suxian No.2’. Moreover, the promoter activity of *Red-AtrDODA1-1pro* in ‘Suxian No. 1’ was higher than that of *Green-AtrDODA1-1pro* in ‘Suxian No.2’. Further analysis revealed that the *Red-AtrDODA1-1pro* sequence contained one more MYB binding site involved in the photoreaction element than *Green-AtrDODA1-1pro* sequence, which might be more conducive to binding with MYB transcription factors under light condition. MYB and WRKY transcription factors were involved in betalain synthesis. *HpWRKY44* directly binds and activates the expression of HpCytP450-like1 by recognizing the W-box element in the promoter, and promotes the biosynthesis of betalain in pitaya [6]. Beet BvMYB1 directly binds to the *BvDODA* and *BvCYP76AD1* promoters to promote their expression, thereby increasing the accumulation of betalains [20]. In our study, through the analysis of potential binding sites of transcription factors on promotors, it was found that the *Red-AtrDODA1-1pro* contained three WRKY binding sites and three MYB binding sites, while *Green-AtrDODA1-1pro* contained one WRKY binding site and one MYB binding site. Whether these differences in promoter sequence affected promoter activity and thus affected betalain synthesis needs further study.

The betalain accumulation in Portulaca genus was promoted with an appropriate concentration of methyl jasmonate [67]. 100 μM MeJA could enhance the accumulation of betalains and other bioactive compounds in *Alternanthera philoxeroides* and *Alternanthera sessiis* [68]. Betalain was enriched in the leaves of *Alternanthera sessiis* with methyl jasmonate treatment for 48 h and 96 h [69]. In this study, both the *Red-AtrDODA1-1pro* and the *Green-AtrDODA1-1pro* responded to the induction of methyl jasmonate and gibberellins. With the extension of treatment time, the induction intensity of *Red-AtrDODA1-1pro* and *Green-AtrDODA1-1pro* in response to methyl jasmonate reached the maximum, indicating that methyl jasmonate promoted the activity of *Red-AtrDODA1-1pro* and *Green-AtrDODA1-1pro*. However, the *Red-AtrDODA1-1pro* and *Green-AtrDODA1-1pro* in response to gibberellin were different. The promoter activity of *Red-AtrDODA1-1pro* showed a trend of inhibition under the induction of gibberellin, whereas promoter activity of *Green-AtrDODA1-1pro* in response to gibberellin reached the maximum at 100 μm-12 h, indicating that gibberellin inhibited the activity of *Red-AtrDODA1-1pro* and promoted the activity of *Green-AtrDODA1-1pro*. It is noteworthy that the maximum GUS enzyme activity of the *Green-AtrDODA1-1pro* of ‘Suxian No.2’ in response to methyl jasmonate and gibberellin was 71 and 104, respectively, implying the inducing effect of gibberellin on the *Green-AtrDODA1-1pro* of ‘Suxian No.2’ might be greater than that of methyl jasmonate. However, the induction effect of methyl jasmonate in the *Red-AtrDODA1-1pro* of ‘Suxian No.1’ was more important.

## 4. Materials and Methods

### 4.1. Materials and Treatments

‘Suxian No.1’ (red amaranth) and ‘Suxian No.2’ (green amaranth) were provided by the Suzhou Academy of Agricultural Sciences. Amaranth seeds were washed with 10 mL of 75% alcohol for 50 s, rinsed with sterile water for 1 min, and soaked in 10% sodium hypochlorite for 18 min for sterilization. Finally, they were rinsed with sterile water for 1 min and this was repeated seven times. The following treatment groups were set up: (1) Sterilized seeds were inoculated into MS medium supplemented with 30 g/L sucrose and 7 g/L agar, and then cultured under three light qualities: white light, blue light, and darkness; (2) Sterilized seeds were inoculated into MS media supplemented with either 1 mg/L gibberellin or 2 mg/L paclobutrazol, without hormones as a control. Each treatment was inoculated into 20 bottles, with 20 seeds per bottle, and cultured in a tissue culture room with 16 h light/8 h dark at 25 °C. After 7 days, seedlings’ roots, stems, and leaves were collected, quickly frozen in liquid nitrogen, and stored at −80 °C for qRT-PCR expression analysis. Each treatment was biologically repeated three times (3). RNA samples of salt stress of ‘Suxian No.1’ were previously stored in the laboratory [62].

*Nicotiana benthamiana* seeds were sown in potting trays and cultured in a constant temperature incubator at 25 °C, 65% humidity with 16 h light/8 h dark. After 25 days of cultivation, they were used for Agrobacterium infection to detect the promoter activity of *AtrDODA1-1*. Meanwhile, the response of *AtrDODA1-1* promoter to the different concentrations (0 μM, 100 μM, 200 μM) of exogenous MeJA (Aladdin Holdings Group Co., Ltd., Beijing, China, N118453) and GA_3_ (Aladdin Holdings Group Co., Ltd., N118453) sprayed on the tobacco leaves was detected. The tobacco leaves were collected at 0 h, 3 h, 6 h, 12 h, and 24 h for analysis.

### 4.2. Bioinformatics Analysis

#### 4.2.1. Identification of DODA Gene Family Members in Amaranth

*Amaranthus tricolor* genomic data were downloaded from http://ftp.agis.org.cn:8888/~fanwei/Amaranthus_tricolor/, (accessed on 1 October 2024). The plot hidden Markov model (HMM) profile of LigB family (PF02900) was downloaded from the Pfam (http://pfam-legacy.xfam.org/, accessed on 1 October 2024) website to identify *DODA* genes from the amaranth genome with an e-value threshold of -5. All candidate members were further identified using CD-Seach in NCBI and PFAM domain in SMART (http://smart.embl-heidelberg.de/, accessed on 1 October 2024). Finally, the *AtrDODA* family members were named according to the annotation and nomenclature of *DODA* family members in Caryophyllales [14].

#### 4.2.2. Structural Characterization of DODA Gene Family Members in Amaranth

GSDS2.0 (http://gsds.cbi.pku.edu.cn/, accessed on 2 October 2024) was used to characterize the introns and exons of *DODA* genes for gene structure and online mapping; the protein conserved motifs of *AtrDODA* were analyzed by MEME (https://meme-suite.org/, accessed on 2 October 2024), and TBtools-II v2.068 software was used for visualization. Each candidate *DODA* gene was further confirmed using SMART (http://smart.embl-heidelberg.de/, accessed on 2 October 2024) and the Conserved Domain Database (CDD, http://www.ncbi.nlm.nih.gov/Structure/cdd/wrpsb.cgi, accessed on 2 October 2024). The chromosomal location of *DODA* genes was analyzed on amaranth genome and visualized with TBtools-II v2.068 software.

#### 4.2.3. Gene Collinearity Analysis and Phylogenetic Tree Construction of DODA

Gene synteny analysis of *DODAs* of amaranth, Beta vulgaris, Pitaya, Bougainvillea, and Portulaca oleracea was performed using MCScanX (https://github.com/wyp1125/MCScanX, accessed on 3 October 2024) using the MUSCLE method. A total of 70 amino acid sequences of *DODA* family for phylogenetic analysis were obtained from the Caryophyllales order. A phylogenetic tree was constructed using the JTT model using the Maximum Likelihood method (ML) using MEGA X with 1000 bootstrap replicates.

#### 4.2.4. Multiple Sequence Alignment Analysis of DODA Amino Acids in Amaranth and Other Species

The homologous DODA proteins in different species were downloaded from NCBI (https://www.ncbi.nlm.nih.gov/, accessed on 3 October 2024). DODA proteins of amaranth and other caryophylla species were analyzed by using DNAMAN 9.0 software.

#### 4.2.5. Prediction of Cis-Acting Elements in the Promoter Sequences of AtrDODA Genes

The 2000 bp promotor region of each *DODA* gene was obtained by using TBtools-IIv2.068 software, then submitted to the PlantCARE (http://bioinformatics.psb.ugent.be/webtools/plantcare/html/, accessed on 5 October 2024) promoter analysis tool to identify potential cis-regulatory elements.

#### 4.2.6. Analysis of Expression Patterns of the AtrDODA Gene Family Based on the Transcriptome Data

The expression patterns of *AtrDODA* family members were analyzed according to TPM values based on the transcriptome data. A heatmap of their expression patterns in amaranth was again visualized using TBtools-II v2.068 software.

#### 4.2.7. RNA Extraction and qRT-PCR Analysis

Total RNA was extracted using a polysaccharide polyphenol plant RNA extraction kit (Yeasen, China), according to the manufacturer’s instructions. The OD value and concentration of RNA were determined using an ultra-micro spectrophotometer (Thermo Electron Corp, Waltham, MA, USA), and its integrity was detected by using a 1% agarose gel electrophoresis.

First-strand cDNA was synthesized by using TransScript One-Step gDNA Removal and cDNA Synthesis SuperMix (TransGen Biotech, Beijing, China) for quantitative real time-PCR (qRT-PCR). qRT-PCR was performed in optical 96-well plates using the Roche LightCycler 480 instrument (Roche, Basel, Sweden). The reaction system was carried out in a 20 μL volume, containing 10 μL of SYBR Premix Ex Taq, 0.8 μL of gene specific primers, 1 μL diluted cDNA, and 8.2 μL of ddH2O. The qRT-PCR reaction procedure was as follows: pre-denaturation at 95 °C for 30 s, 45 cycles of denaturation at 95 °C for 10 s, and annealing/extension at 58 °C for 20 s, followed by at 72 °C for 12 s. Three biological repeats were performed for each material. SAND was used as the internal reference gene. The relative expression was calculated according to the 2^−ΔΔCt^ method. The data were statistically analyzed using Excel 2019 software, and significance analysis (*p* < 0.05) was performed using the Duncan method in SPSS 24 software and plotted using Origin 2017 software. The primer pairs used for the qRT-PCR analysis of *DODA* genes are shown in Appendix A.

#### 4.2.8. Determination of Betalain Content in Amaranth

Referring to Daryl’s [67] investigation, 0.3g of amaranth material was placed in a mortar and ground to powder with liquid nitrogen. Then, 0.05 g powder was placed into five 2 mL centrifuge tubes. A total of 1.5 mL of 95% anhydrous ethanol was added to each centrifuge tube, vortexed for 30 s, mixed well, and stood in a 4 °C refrigerator for 30 min. The tubes were placed in a centrifuge at 10,000 r·min-1, 4 °C, and centrifuged for 12 min; the supernatant was then discarded. A total of 1.5 mL of deionized water was added, vortexed for 30 s, mixed well, and stood in a 4 °C refrigerator for another 30 min. It was centrifuged again at 10,000 r·min-1, 4 °C, for 12 min, and the supernatant was removed. A UV spectrophotometer (UV-900, Shanghai Yuanxi Instrument Co., Ltd., Shanghai, China) was used to measure the absorbance of betacyanin and betaxanthin at 538 nm and 465 nm, respectively, with three biological replicates. The content of betalains was calculated according to the formula:$$\mathrm{Betalain}\;\mathrm{content}(\mathrm{mg}/g)=\frac{[(A\times DF\times M\times 1000)/(\varepsilon \times L)]\times V}{m\times 1000}$$

A represents the absorbance of betalain; DF represents the dilution factor; M represents the relative molecular mass (with M_betacyanin_ = 550  g/mol and M_betaxanthin_ = 308  g/mol); ε represents the molar absorptivity (with ε_betacyanin_ = 60,000  L/(mol.cm) and ε_betaxanthin_ = 48,000  L/(mol.cm); L represents the thickness of the cuvette in centimeters; V represents the total volume of the extract solution in milliliters; m represents the mass of the sample in grams.

#### 4.2.9. Analysis of the Subcellular Localization of the Amaranth AtrDODA Proteins

The coding sequences of the *AtrDODA1-1*, *AtrDODA2-1*, and *AtrDODA1-2* genes were inserted into the pRI101-AN-eGFP vector to form recombinant plasmid vectors (the primer sequences are listed in Appendix A). The pRI101-*35S*::eGFP empty vector and pRI101-*35S*::AtrDODAs-eGFP were transformed into GV3101 *Agrobacterium* competent cells,. *Agrobacterium* strain GV3101 cells were infiltrated into onion inner epidermis cells. After culturing in the dark at 28 °C for 2 days, transient expression of eGFP was recorded using an Olympus model FV1200 4laser confocal microscope (Shibuya, Japan) (laser) at 405 nm. To evaluate nuclear localization, the onion epidermis cells were stained with DAPI, and the fluorescence signal was visualized by laser confocal microscopy at 473 nm.

#### 4.2.10. Analysis of AtrDODA1-1 Promoter Activity and Response to Exogenous Hormones

The promoter sequence of *Red-AtrDODA1-1pro* and *Green-AtrDODA1-1pro* was cloned from ‘Suxian No.1’ and ‘Suxian No.2’ gDNA, respectively. The two promoter sequences were constructed into the pCAMBIA1301 vector to form *pRed-AtrDODA1-1pro*::GUS and *pGreen-AtrDODA1-1pro*::GUS, respectively, by using a one-step rapid cloning kit (10911ES20,Yeasen, China). Subsequently, the recombinant plasmids and the empty vector pCAMBIA1301 were transformed into GV3101 *Agrobacterium* competent cells. *Agrobacterium* strain GV3101 cells were infiltrated into lower epidermal cells of the tobacco leaves and repeated for 3 plants. After injection, the tobacco was placed under dark conditions at 25 °C for 1 day, followed by growth under light conditions for 2 days. The tobacco leaves transformed with Agrobacterium containing *p35S*::GUS, *pRed-AtrDODA1-1pro*::GUS, or *pGreen-AtrDODA1-1pro*::GUS constructs were subjected to GUS histochemical staining, and then GUS enzyme activity determination by using Image J software. The relative expression level of the GUS gene was analyzed by using qRT-PCR.

Meanwhile, the different concentrations (0 μM, 100 μM, 200 μM) of exogenous MeJA and GA_3_ was sprayed on the tobacco leaves transformed by the *pRed-AtrDODA1-1pro*::GUS and *pGreen-AtrDODA1-1pro*::GUS on the 2nd day of cultivation. Subsequently, the tobacco leaves were collected for GUS staining and enzyme activity determination by using Image J software at 0 h, 3 h, 6 h, 12 h, and 24 h after the exogenous treatment.

The GUS staining solution consisted of sodium phosphate buffer at pH = 7, 0.25 M Na_2_EDTA·2H_2_O, 50 mM K_3_Fe(CN)_6_, 50 mM K_4_Fe(CN)_6_, 0.1% Triton X-100, and 100 mM X-Gluc. The tobacco leaves were placed in centrifuge tubes and submerged with an appropriate amount of GUS staining solution to cover the plant material. They were incubated at 37 °C for 12 h, then the chlorophyll was de-stained with 70% ethanol at 37 °C for 1.5 h, replacing the de-staining solution 2–3 times during this period, and the de-staining process was repeated until the green color completely faded. After de-staining, the samples were stored in 75% ethanol at 4 °C for observation and photographed. The GUS staining enzyme activity was effectively evaluated by using Image J (Version 1.54g, NIH) software. The relative expression level of the *GUS* gene was analyzed by using qRT-PCR.

## 5. Conclusions

In summary, three *AtrDODA* family members were identified based on the amaranth genome and transcriptome. Subcellular localization showed that the three AtrDODA members were located in the nucleus. The betacyanin and betaxanthin were promoted by paclobutrazol treatment in the leaves and stems of ‘Suxian No.1’ (red), while they were inhibited by gibberellin and darkness. qRT-PCR showed that the expression of *AtrDODA1-1* in betalain synthesis was different from that of *AtrDODA1-2* and *AtrDODA2-1*. The activity of *Red-AtrDODA1-1pro* of ‘Suxian No.1’ was higher than that of *Green-AtrDODA1-1pro* of ‘Suxian No.2’. *AtrDODA1-1pro* responded to the induction of MeJA and GA_3_, and the activity of the ‘Suxian No.1’ *Red-AtrDODA1-1pro* was significantly higher than that of the ‘Suxian No.2’ *Green-AtrDODA1-1pro*. *AtrDODA1-1pro* responded to the induction of MeJA and GA_3_.

## Figures and Tables

**Figure 1 plants-14-00454-f001:**
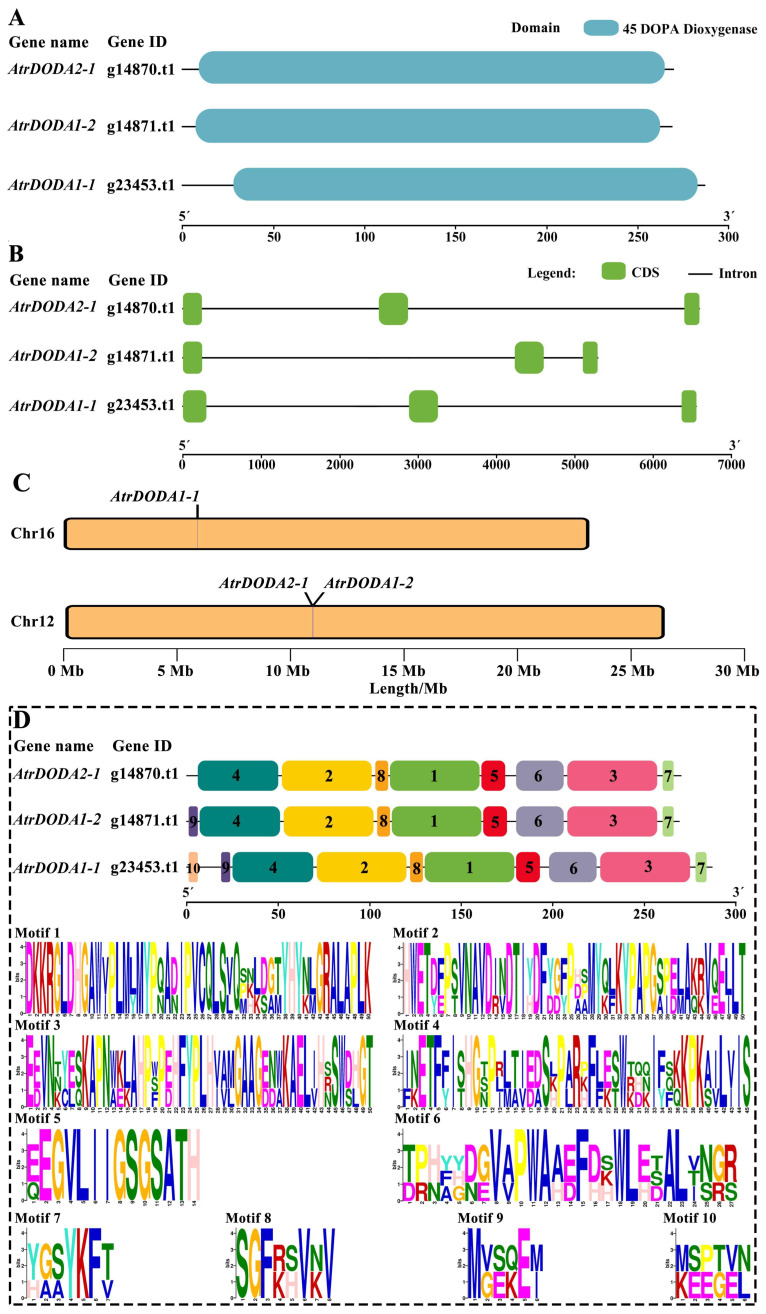
Structure of the *AtrDODA* gene, domain, motif, and chromosome localization in amaranthus: (**A**) Conserved domains of AtrDODAs; (**B**) Gene structure of *AtrDODAs*; (**C**) Chromosome localization of *AtrDODAs* genes; (**D**) Conserved motifs of AtrDODAs.

**Figure 2 plants-14-00454-f002:**
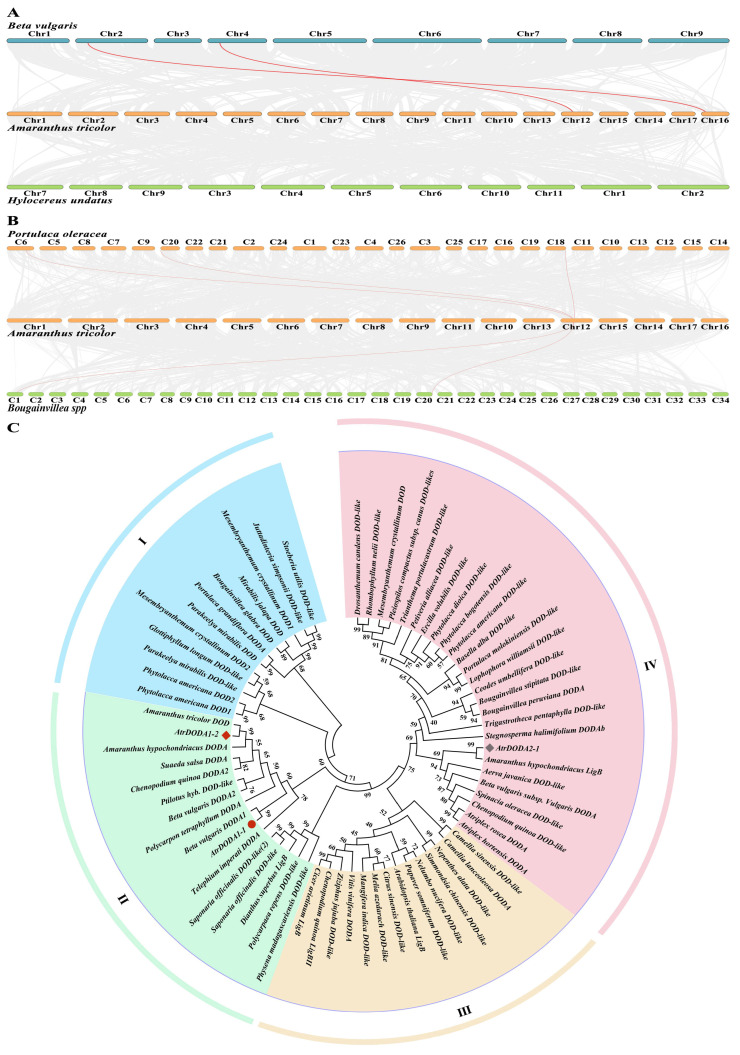
Collinearity and phylogenetic tree of *AtrDODA* in amaranth: (**A**) Interspecific collinearity analysis of *DODA* between *Amaranthus tricolor* L., *Beta vulgaris*, and *Hylocereus undatus*. (**B**) Interspecific collinearity analysis of DODA between *Amaranthus tricolor*, *Portulaca oleracea*, and *Bougainvillea*. (**C**) Phylogenetic tree of *DODA* homologs between amaranth and other plants. The roman letters and different colors represents the four different groups.

**Figure 3 plants-14-00454-f003:**
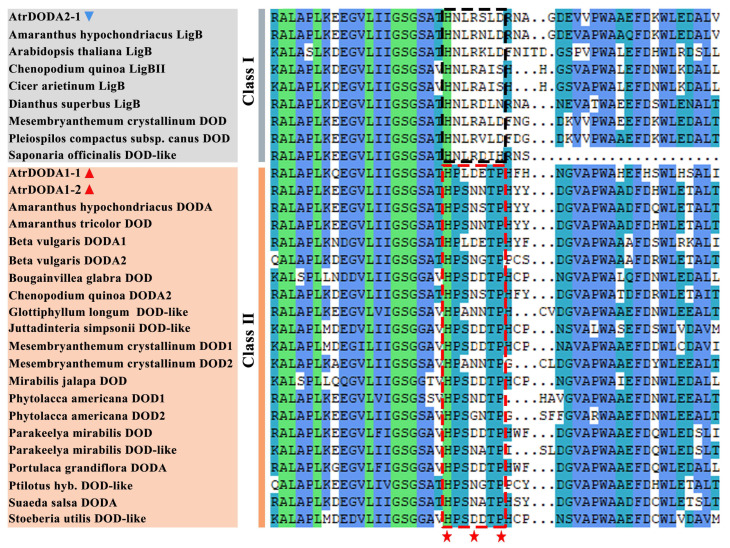
Multiple sequence alignment of DODA amino acids in amaranth and other species: Blue triangles represent Class I members of AtrDODA, and red triangles represent Class II members of AtrDODA. Asterisks indicate acidic residues that may be important for DOD activity/DODA recognition.

**Figure 4 plants-14-00454-f004:**
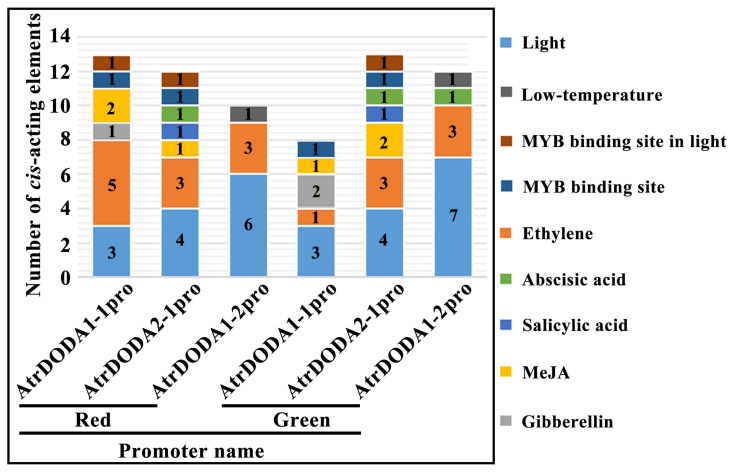
Analysis of *cis*-acting elements in the promoter of *AtrDODA*. The numbers in the figure represent the counts of various *cis*-acting elements, with red indicating ‘Suxian No.1’ and green indicating ‘Suxian No.2’.

**Figure 5 plants-14-00454-f005:**
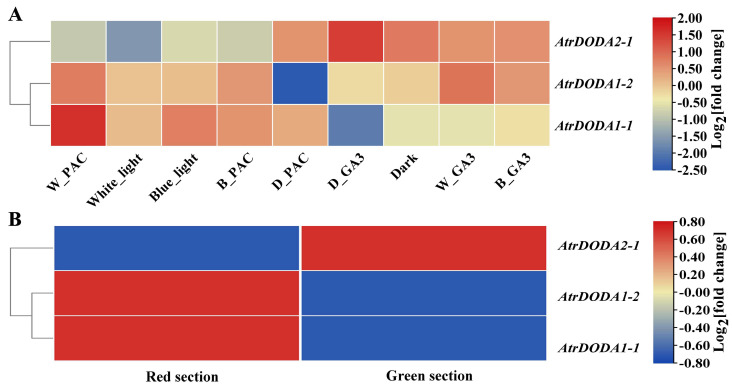
Expression heatmap of *AtrDODA*: (**A**) Gene expression heatmap of *AtrDODA* in ‘Suxian No.1’ (red) under blue and white light quality, dark treatment, and plant hormone treatment; (**B**) Heatmap of *AtrDODA* gene expression in the red and green sections of the leaves of red amaranth.

**Figure 6 plants-14-00454-f006:**
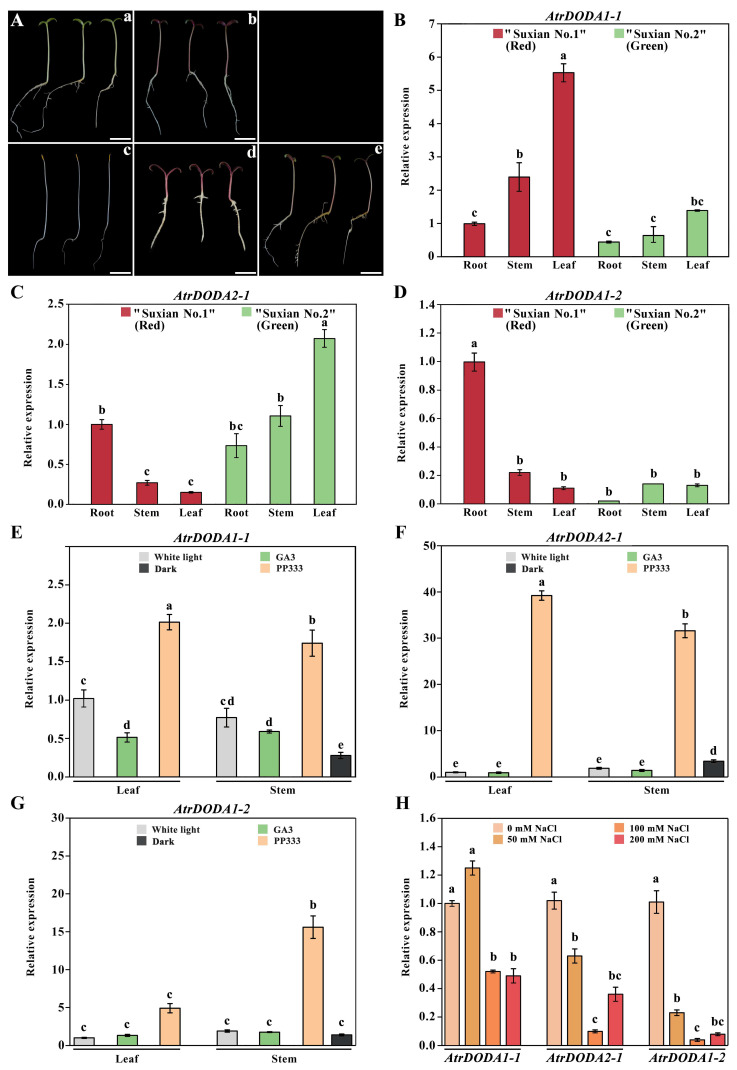
Expression patterns of *AtrDODA* genes in amaranth: (**A**) Phenotype images of different varieties and ‘Suxian No.1’ (red) amaranth under various treatments. (**a**): ‘Suxian No.2’ under white light treatment; (**b**): ‘Suxian No.1’ under white light treatment; (**c**): ‘Suxian No.1’ under dark treatment; (**d**): ‘Suxian No.1’ treated with 2 mg/L paclobutrazol; (**e**): ‘Suxian No.1’ treated with 1 mg/L gibberellin. (**B**) *AtrDODA1-1*, (**C**) *AtrDODA2-1*, and (**D**) *AtrDODA1-2* expression patterns in ‘Suxian No.1’ and ‘Suxian No.2’ under white light treatment. (**E**) *AtrDODA1-1*, (**F**) *AtrDODA2-1*, and (**G**) *AtrDODA1-2* expression patterns in ‘Suxian No.1’ under white light, gibberellin, paclobutrazol, and dark treatments. (**H**) Expression patterns of *AtrDODAs* genes under different concentrations of salt treatment. The scale bar in A represents 1 cm, and different lowercase letters in (**B**–**H**) represent significant differences in gene expression (*p* < 0.05).

**Figure 7 plants-14-00454-f007:**
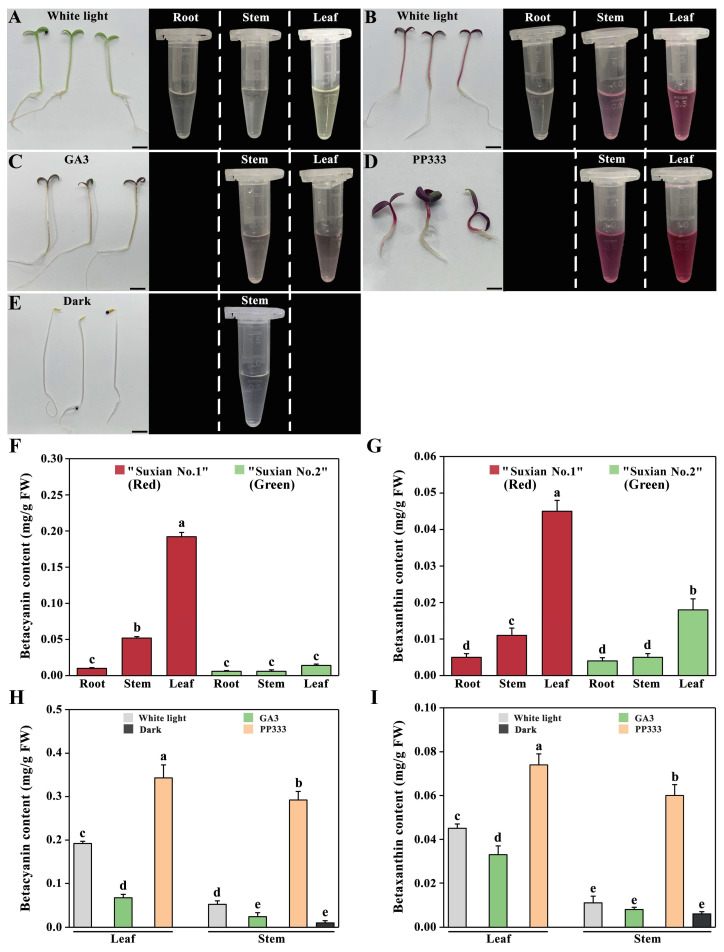
Phenotypes and betacyanin content in amaranth: (**A**) Betacyanin samples from ‘Suxian No.2’ (green) amaranth under white light treatment; (**B**) Betacyanin samples from ‘Suxian No.1’ (red) amaranth under white light treatment, (**C**) under gibberellin treatment, (**D**) under paclobutrazol treatment, (**E**) and under dark treatment; (**F**) Betacyanin content in roots, stems, and leaves of ‘Suxian No.1’ (red) and ‘Suxian No.2’ (green) amaranth; (**G**) Betaxanthin content in roots, stems, and leaves of ‘Suxian No.1’ (red) and ‘Suxian No.2’ (green) amaranth; (**H**) Betacyanin content and (**I**) betaxanthin content in ‘Suxian No.1’ (red) amaranth under different light qualities, gibberellin, and paclobutrazol treatments. The scale bar in (**A**–**E**) is 1 cm, and different lowercase letters in (**F**–**I**) represent significant differences in betacyanin and betaxanthin content (*p* < 0.05).

**Figure 8 plants-14-00454-f008:**
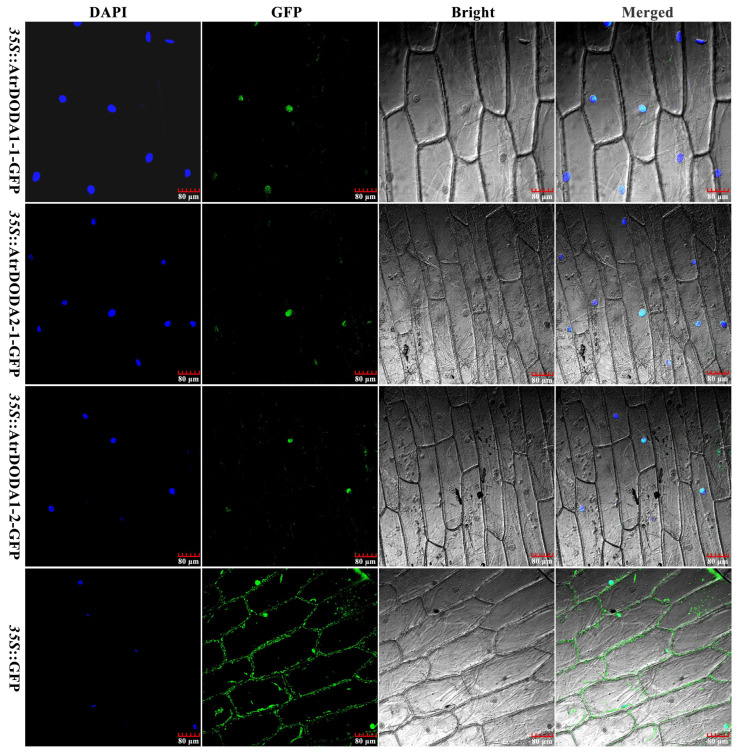
Subcellular localization of AtrDODAs. DAPI panel represents green fluorescence co-localized with DAPI staining; GFP panel represents GFP fluorescence; Bright panel represents bright field photograph; Merge panel represents bright field overlay of GFP fluorescence.

**Figure 9 plants-14-00454-f009:**
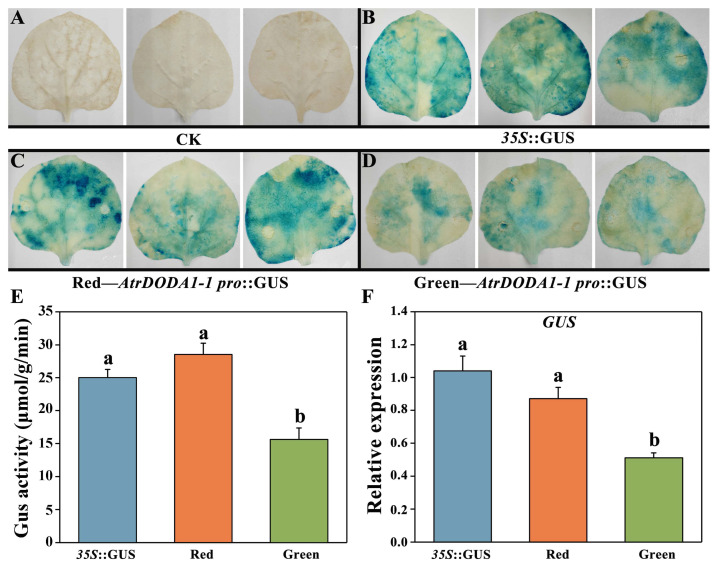
Analysis of promoter activity of AtrDODA1-1 in red and green amaranth: (**A**) GUS staining of wild-type control tobacco; (**B**) GUS staining of tobacco with empty vector pCAMBIA1301; (**C**) GUS staining of tobacco with the promoter of red AtrDODA1-1; (**D**) GUS staining of tobacco with the promoter of green AtrDODA1-1; (**E**) β-Glucuronidase (GUS) enzyme activity; (**F**) Expression of the GUS gene. The different lowercase letters in (**E**,**F**) represent significant differences (*p* < 0.05).

**Figure 10 plants-14-00454-f010:**
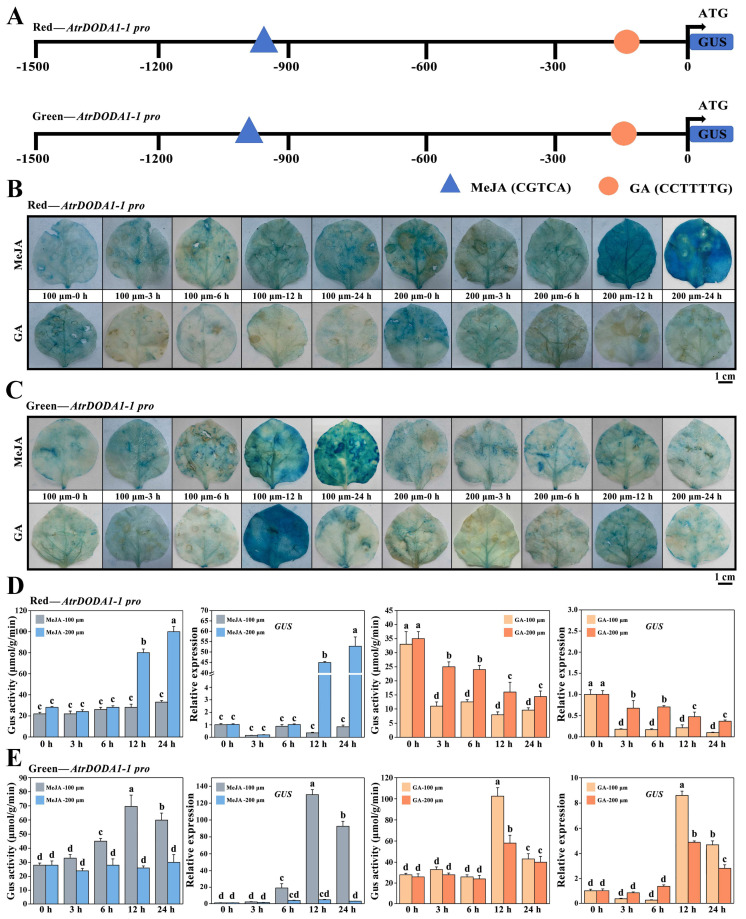
Analysis of the *AtrDODA1-1* promoters in red and green amaranth in response to exogenous hormone treatments: (**A**) Construction of *GUS* expression vectors for the *AtrDODA1-1* promoters of red and green amaranth; (**B**) GUS staining of tobacco with the red *AtrDODA1-1* promoter under MeJA and GA treatments; (**C**) GUS staining of tobacco with the *green AtrDODA1-1* promoter under MeJA and GA treatments; (**D**) β-Glucuronidase (GUS) enzyme activity and *GUS* gene expression of the *red AtrDODA1-1* promoter under MeJA and GA treatments; (**E**) β-Glucuronidase (GUS) enzyme activity and *GUS* gene expression of the *green AtrDODA1-1 promoter* under MeJA and GA treatments. In (**D**,**E**), different lowercase letters represent significant differences in gene expression (*p* < 0.05).

## Data Availability

All datasets generated for this study are included in the article.

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
