# Peer review of "Differential Expression of Amaranth AtrDODA Gene Family Members in Betalain Synthesis and Functional Analysis of AtrDODA1-1 Promoter"

_plants, 2025, doi:10.3390/plants14030454_

Round 1

Reviewer 1 Report

Comments and Suggestions for Authors

The article examines the roles of AtrDODA gene family members in betalain synthesis in Amaranthus tricolor. Below is an academic and scientific review emphasizing strengths and weaknesses. The study uses advanced bioinformatics tools for identifying AtrDODA genes and analyzing their structural characteristics, conserved motifs, and chromosomal localization. It integrates promoter activity analysis, gene expression profiling, and betalain quantification, offering a robust multi-disciplinary approach. Below, the key weaknesses and areas for enhancement are outlined.

Lack of Functional Validation: While the study links AtrDODA gene expression and promoter activity to betalain synthesis, direct functional validation is missing. Techniques such as CRISPR-Cas9 gene editing or RNAi-mediated gene silencing in Amaranthus or model plants (e.g., Nicotiana benthamiana or Arabidopsis thaliana) could have been used to directly confirm the roles of AtrDODA1-1, AtrDODA1-2, and AtrDODA2-1. This gap weakens the causative claims made about gene activity and betalain production.

Limited Genetic Context and Comparative Insight: The study compares AtrDODA genes primarily with closely related species like Beta vulgaris, Hylocereus undatus, and Portulaca oleracea. While this provides some phylogenetic perspective, broader comparisons with a more diverse set of Caryophyllales species (e.g., Mirabilis jalapa or Bougainvillea spp.) would enrich the evolutionary context. Furthermore, the study does not explore whether AtrDODA genes underwent specific selective pressures that might explain their functional divergence.

Superficial Promoter Analysis: The paper identifies sequence differences in AtrDODA1-1 promoters between red and green Amaranthus varieties but does not delve into how these variations affect transcription factor binding or regulatory efficiency. For instance, chromatin immunoprecipitation (ChIP) assays or electrophoretic mobility shift assays (EMSAs) could validate the interaction between identified transcription factors (e.g., MYB and WRKY) and the promoter regions. Without this, the link between promoter sequence variations and betalain content remains speculative.

Simplified Methodology for Betalain Quantification: The study uses UV spectrophotometry to measure betalain content, which is not highly specific and may lead to inaccuracies due to interference from other pigments or compounds. High-performance liquid chromatography (HPLC) or liquid chromatography-mass spectrometry (LC-MS) could provide a detailed profile of betalain derivatives (e.g., betacyanins vs. betaxanthins), improving the robustness of the data.

Incomplete Exploration of Environmental Factors: While light and hormones are investigated, other critical environmental factors like temperature, water availability, and soil nutrients, which are known to influence betalain synthesis, are not explored. Additionally, the interactions between these factors and hormonal or genetic regulation remain unexamined. Such omissions limit the applicability of the findings to agricultural contexts, especially in variable or stressful environments.

Inconsistent or Underexplained Findings: The study reports that dark conditions significantly inhibit betalain synthesis, but differences in light quality (white vs. blue) do not affect synthesis substantially. This contradicts other findings in Caryophyllales plants, where blue light often promotes betalain accumulation. The absence of mechanistic explanations or comparisons with existing research raises questions about experimental consistency.

Additionally, while AtrDODA2-1 expression is highlighted as critical in the green Amaranthus variety, its precise biochemical role in betaxanthin synthesis is not explored in depth. Enzyme activity assays, for instance, could elucidate its specific function.

Limited Real-World Applicability: Although the study has potential agricultural applications, it does not evaluate betalain production under field conditions. Factors like soil composition, pest resistance, or natural stressors, which significantly impact crop yield and pigment accumulation, are ignored. The controlled laboratory settings do not necessarily reflect real-world conditions, reducing the practical value of the findings.

Overemphasis on Correlation: Much of the study relies on correlations between gene expression and betalain content. While these correlations are valuable, they do not establish causation. The absence of functional assays or validation experiments (e.g., enzyme assays, genetic transformations) weakens the evidence for the proposed mechanisms.

Reproducibility Concerns: The study mentions "unpublished findings" (e.g., hormonal induction of betalains in roots) to support its conclusions. Such claims, without data or citations, undermine scientific rigor. All experimental findings need to be fully documented and published to ensure transparency and reproducibility.

Author Response

          Thank you so much for reviewing our manuscript entitled "Differential expression of amaranth AtrDODA gene family members in betalains synthesis and functional analysis of AtrDODA1-1 promoter". We also deeply appreciate you for your critical review of the manuscript with thoughtful and constructive comments, based on which we have revised the manuscript.

1. Lack of Functional Validation: While the study links AtrDODA gene expression and promoter activity to betalain synthesis, direct functional validation is missing. Techniques such as CRISPR-Cas9 gene editing or RNAi-mediated gene silencing in Amaranthus or model plants (e.g., Nicotiana benthamiana or Arabidopsis thaliana) could have been used to directly confirm the roles of AtrDODA1-1, AtrDODA1-2, and AtrDODA2-1. This gap weakens the causative claims made about gene activity and betalain production.

Response: Thank you for your valuable suggestion. CRISPR-Cas9 gene editing or RNAi-mediated gene silencing is a powerful tool to verify gene functions. However, so far a stable genetic transformation system of Amaranthus tricolor has not been established in both peer laboratories and our research group.. Our previous studies have shown that AtrDODA gene was involved in betalain synthesis in Amaranthus tricolor. The analysis of AtrDODA gene expression and promoter activity were performed to verify preliminarily its function. We will continue to further perform functional verification experiments to confirm the roles of AtrDODA1-1, AtrDODA1-2, and AtrDODA2-1, and provide more solid evidence.

2. Limited Genetic Context and Comparative Insight: The study compares AtrDODA genes primarily with closely related species like Beta vulgaris, Hylocereus undatus, and Portulaca oleracea. While this provides some phylogenetic perspective, broader comparisons with a more diverse set of Caryophyllales species (e.g., Mirabilis jalapa or Bougainvillea spp.) would enrich the evolutionary context. Furthermore, the study does not explore whether AtrDODA genes underwent specific selective pressures that might explain their functional divergence.

Response: Thank you for your valuable suggestion. We have collected the sequence data of AtrDODA homologous genes from multiple other Caryophyllales species, including Mirabilis jalapa and Bougainvillea spp. Using phylogenetic analysis software. In addition, with the help of the bioinformatics tool KaKs_Calculator, the Ka/Ks ratio can be accurately estimated to assess whether the Amaranthus tricolor AtrDODA gene is under specific selective pressures and thus infer its functional divergence. We have modified the section (Shown in Figure 2B and 2C, Supplementary Table 2, and showed in lines 166-179, 732-733)

3.Superficial Promoter Analysis: The paper identifies sequence differences in AtrDODA1-1 promoters between red and green Amaranthus varieties but does not delve into how these variations affect transcription factor binding or regulatory efficiency. For instance, chromatin immunoprecipitation (ChIP) assays or electrophoretic mobility shift assays (EMSAs) could validate the interaction between identified transcription factors (e.g., MYB and WRKY) and the promoter regions. Without this, the link between promoter sequence variations and betalain content remains speculative.

Response: Thank you for your valuable suggestion. In the present study, our results showed that the activity of the Red-AtrDODA1-1pro promoter in 'Suxian No.1' was higher than that of the Green-AtrDODA1-1pro promoter in 'Suxian No.2' by using GUS staining experiments and exogenous hormone treatment experiments. Moreover, through the analysis of the potential binding sites of transcription factors on the promoters, we found that the number of WRKY binding sites and MYB binding sites are different between the Red-AtrDODA1-1pro promoter and the Green-AtrDODA1-1pro promoter. The natural variation in cis-regulatory regions usually affects the plant phenotypes by altering gene expression according to the literatures (Showed in lines 510 - 525). We will further delve into how these variations affect transcription factor binding or regulatory efficiency in the future.

4. Simplified Methodology for Betalain Quantification: The study uses UV spectrophotometry to measure betalain content, which is not highly specific and may lead to inaccuracies due to interference from other pigments or compounds. High-performance liquid chromatography (HPLC) or liquid chromatography-mass spectrometry (LC-MS) could provide a detailed profile of betalain derivatives (e.g., betacyanins vs. betaxanthins), improving the robustness of the data.

Response: Our determination method of the betalain content is feasible because we cited the literature[Cao S , Liu T , Jiang Y ,et al. The effects of host defence elicitors on betacyanin accumulation in Amaranthus mangostanus seedlings[J].Food Chemistry, 2012, 134(4):1715-1718] and our published articles. Moreover, your advice is important, and we will improve the determination method, such as High-performance liquid chromatography (HPLC) or Liquid chromatography-mass spectrometry (LC-MS), to provide a detailed profile of betalain derivatives.

5. Incomplete Exploration of Environmental Factors: While light and hormones are investigated, other critical environmental factors like temperature, water availability, and soil nutrients, which are known to influence betalain synthesis, are not explored. Additionally, the interactions between these factors and hormonal or genetic regulation remain unexamined. Such omissions limit the applicability of the findings to agricultural contexts, especially in variable or stressful environments.

Response: Thank you for your suggestion. There are many environmental factors affecting the betalain synthesis. Light and hormones are important environmental factors that regulate plant growth, development and quality. We carried out the study that light and hormones regulated betalain synthesis according to our previous research. Therefore, we focused on the study that the light and hormones regulated the expression of the AtrDODA gene and the betalain synthesis. In order to elaborate on the individual and interactive mechanisms of other environmental factors, we will future explore the research to apply to agricultural production.Thank you for your valuable suggestion..

6. Inconsistent or Underexplained Findings: The study reports that dark conditions significantly inhibit betalain synthesis, but differences in light quality (white vs. blue) do not affect synthesis substantially. This contradicts other findings in Caryophyllales plants, where blue light often promotes betalain accumulation. The absence of mechanistic explanations or comparisons with existing research raises questions about experimental consistency.

Response: We apologize for some seemingly inconsistent results. We corrected it in Figures 6-7 (Showed in lines 259-262, 268-269, 268-269, and 311-312).

7. Additionally, while AtrDODA2-1 expression is highlighted as critical in the green Amaranthus variety, its precise biochemical role in betaxanthin synthesis is not explored in depth. Enzyme activity assays, for instance, could elucidate its specific function.

Response: We sincerely thank you for your profound insights. We performed qRT-PCR assay and analysis of promoter activity of AtrDODA2-1 to exploreits function. These results showed the gene has a certain function in the manuscript. However, we will explore in depth its specific function through more experiments in future.

8. Limited Real-World Applicability: Although the study has potential agricultural applications, it does not evaluate betalain production under field conditions. Factors like soil composition, pest resistance, or natural stressors, which significantly impact crop yield and pigment accumulation, are ignored. The controlled laboratory settings do not necessarily reflect real-world conditions, reducing the practical value of the findings.

Response: We sincerely appreciate your profound comments.. There are many environmental factors affecting the betalain synthesis. Light and hormones are important environmental factors that regulate plant growth, development and quality. We focused on the potential mechanism of light and hormone influence on betalain synthesis in the manuscript. The result is of great significance and application value for the factory production of amaranth plants. We will evaluate betalain production under field conditions.

9. Overemphasis on Correlation: Much of the study relies on correlations between gene expression and betalain content. While these correlations are valuable, they do not establish causation. The absence of functional assays or validation experiments (e.g., enzyme assays, genetic transformations) weakens the evidence for the proposed mechanisms.

Response: We sincerely appreciate your valuable suggestion. qRT-PCR and promoter activity Analysis are important techniques for preliminary verification of gene function.So we performed the analysis of AtrDODA gene expression and promoter activity to verify preliminarily its function. And the result showed that there is a the correlations between gene expression and betalain content. Our previous studies confirmed that the correlation between gene expression and betalain content was reliable. However, we will further carry out the more experiments to verify the function.

10. Reproducibility Concerns: The study mentions "unpublished findings" (e.g., hormonal induction of betalains in roots) to support its conclusions. Such claims, without data or citations, undermine scientific rigor. All experimental findings need to be fully documented and published to ensure transparency and reproducibility.

Response: We sincerely appreciate your valuable suggestion. We have already corrected it. (Showed in lines 439-442)

           We hope you will agree that we have addressed all the concerns raised in your previous review of the manuscript and improved the manuscript. Thank you for your re-consideration of our manuscript and we are looking forward to your favorable decision.

Reviewer 2 Report

Comments and Suggestions for Authors

Betalains as an unusual class of pigments found in the angiosperm Carophyllales order, synthesis from tyrosine. In recent years, numerous studies have shed light on the evolution of betalain biosynthesis in Carophyllales by generating omicscale of data in phylogenetic and synthetic biology due to their distinct roles in plant pollination by attracting pollinator and stress tolerance. Long ago heterologous expression of DODA gene studied in many species including A. muscaria and Antirrhinum majus, Amaranthus, B. vulgaris and other. Interestingly, characterized enzymes exhibiting DODA activity fall into the LigB gene family that present across all the major kingdoms of land plants. However, the novelty of the is that here author identified screened out three AtrDODA family gene members associated with betalains synthesis in Amaranthus tricolor. Here author showed an excellent effort to prove the hypothesis by using multi -omic strategies.  Moreover, author showed that AtrDODA1-1 respond to plant hormone, which has wider application in plant biology.  The experimental design of the study is perfect and the presentation of the fact is excellent.  However, author suggested to check typos or grammatical errors if there is any.

Author Response

          Betalains as an unusual class of pigments found in the angiosperm Carophyllales order, synthesis from tyrosine. In recent years, numerous studies have shed light on the evolution of betalain biosynthesis in Carophyllales by generating omicscale of data in phylogenetic and synthetic biology due to their distinct roles in plant pollination by attracting pollinator and stress tolerance. Long ago heterologous expression of DODA gene studied in many species including A. muscaria and Antirrhinum majus, Amaranthus, B. vulgaris and other. Interestingly, characterized enzymes exhibiting DODA activity fall into the LigB gene family that present across all the major kingdoms of land plants. However, the novelty of the is that here author identified screened out three AtrDODA family gene members associated with betalains synthesis in Amaranthus tricolor. Here author showed an excellent effort to prove the hypothesis by using multi -omic strategies.  Moreover, author showed that AtrDODA1-1 respond to plant hormone, which has wider application in plant biology.  The experimental design of the study is perfect and the presentation of the fact is excellent. However, author suggested to check typos or grammatical errors if there is any.

Response:Thank you so much for reviewing our manuscript entitled "Differential expression of amaranth AtrDODA gene family members in betalains synthesis and functional analysis of AtrDODA1-1 promoter". We also deeply appreciate you for your critical review of the manuscript with thoughtful and constructive comments, based on which we have revised the manuscript.  Thanks for your suggestion. We have corrected the typos or grammatical errors.

Reviewer 3 Report

Comments and Suggestions for Authors

This manuscript titled “Differential Expression of Amaranth AtrDODA Gene Family Members in Betalains Synthesis and Functional Analysis of AtrDODA1-1 Promoter” provides vital information on the DODA gene family specially expression analysis and promoter activity. There are several shortcomings to make the manuscript more fruitful.

Major remarks:

# Line 307: Generally promoter:GUS construct used to observe the tissue specific expression. Author used the construct for measuring the promoter activity in tobacco leaf. Could you please give the more reference and validate the experiment through Promoter:LUC construct in protoplast?

Minor remarks:

#Line 121 and so on: The author could rename the gene. Usually, gene name should be serially named according to the position on chromosome.

#Line 145, Figure 1C: Intron Phase, should be analyzed and described in the manuscript.

#Line 162, 164, Figure 2A, B and so on: Scientific name should be in italic format throughout the manuscript. Collinearity analysis only done in dicotyledonous, author can add monocot and merged all species in a figure or split monocot in figure A and dicot in figure B in response of Amarant.

# Line 173, 594, Figure 2C: Could you please add bootstrap in the node and discuss it in the manuscript? Author can also make categories of the bootstrap like: A high bootstrap value suggests strong support for the grouping at that node, while a lower value indicates less confidence in the grouping. Also need to provide information about the sequence alignment model used clustalW or MUSCLE?

# Line 186: Acidic residues (HPLDETP) for DOD activity /DOPA recognition is belongs to any motif? Please describe in the manuscript too if it belongs to any motif in Figure 1D.

#Line 214, Figure 4: Is it possible to group as stress, hormone, biological etc? Author can add this one in the supplementary file.

#Lines 330, and so on: Please write “cis-regulatory elements” instead of “cis- regulatory element” throughout the manuscript.

Author Response

        Thank you so much for reviewing our manuscript entitled "Differential expression of amaranth AtrDODA gene family members in betalains synthesis and functional analysis of AtrDODA1-1 promoter". We also deeply appreciate you for your critical review of the manuscript with thoughtful and constructive comments, based on which we have revised the manuscript.

1. #Line 307: Generally promoter:GUS construct used to observe the tissue specific expression. Author used the construct for measuring the promoter activity in tobacco leaf. Could you please give the more reference and validate the experiment through Promoter:LUC construct in protoplast?

Response: We sincerely appreciate your profound suggestion. We performed the experiment referring to the article [Evaluation of novel promoters for vascular tissue - specific gene expression in Populus] published by Yi An et al in the journal [Plant Science] in 2024. The researchers constructed three gene promoters into vectors containing the GUS reporter gene. Transgenic plants were obtained through Agrobacterium transformation using these constructs to study and confirm the activity of the promoters in transgenic poplar plants, which provided a valuable reference for us to apply this technique in tobacco leaves.

2. #Line 121 and so on: The author could rename the gene. Usually, gene name should be serially named according to the position on chromosome.

Response: Thanks for your suggestion. We renamed the AtrDODA gene in Amaranthus tricolor referring to the DODA of the Amaranthus genus and the NCBI homologous gene annotations. This will enhance the clarity and consistency of the gene nomenclature in our manuscript. According to BLAST analysis in NCBI, AtrDODA1-1 (localized on chromosome 16) and AtrDODA1-2 (localize on chromosome 12) are homologous to DODA1 of the Amaranthus genus, but AtrDODA2-1 (localized on chromosome 12) is homologous to the LigB family of the Amaranthus genus. Moreover, the rename is also consistent with the classification of the multiple amino acid sequence alignment of DODA in different species. Therefore, our rename method is feasible.

3. Line 145, Figure 1C: Intron Phase, should be analyzed and described in the manuscript.

Response: Thanks for your suggestion. The structure of the AtrDODA gene was shown in the Figure 1B. The gene structure of the three members of the AtrDODA gene family is relatively consistent because each member consists of three exons and two introns. (Showed in lines 129-131)

4. #Line 162, 164, Figure 2A, B and so on: Scientific name should be in italic format throughout the manuscript. Collinearity analysis only done in dicotyledonous, author can add monocot and merged all species in a figure or split monocot in figure A and dicot in figure B in response of Amarant.

Response: Thanks for your suggestion. We have already corrected the all scientific names in whole manuscript. Because betalains are currently found only in some plants of the Caryophyllales order and the fungus Amanita muscaria. All the Caryophyllales plants so far are found in dicotyledonous plants. So it is impossible to select relevant monocotyledonous plant species to conduct the collinearity analysis with Amaranthus plants or other plants containing betalain.

5. # Line 173, 594, Figure 2C: Could you please add bootstrap in the node and discuss it in the manuscript? Author can also make categories of the bootstrap like: A high bootstrap value suggests strong support for the grouping at that node, while a lower value indicates less confidence in the grouping. Also need to provide information about the sequence alignment model used clustalW or MUSCLE?

Response: Thanks for your suggestion. We have modified the Figure 2C, and discuss it (Lines 391-396 and lines 401-402). Meanwhile, we used MUSCLE for sequence alignment (Lines 612-615).

6. Line 186: Acidic residues (HPLDETP) for DOD activity /DOPA recognition is belongs to any motif? Please describe in the manuscript too if it belongs to any motif in Figure 1D.

Response: Thanks for your suggestion. The acidic residues (HPLDETP) used for DOD activity/DOPA recognition in this study were classified referring to the literature [Chung H, Schwinn K E, Ngo H M, et al. Characterisation of betalain biosynthesis in Parakeelya flowers identifies the key biosynthetic gene DOD as belonging to an expanded LigB gene family that is conserved in betalain-producing species[J]. Frontiers in Plant Science. 2015, 6], and the conserved motif analysis was carried out in Figure 1D. The results showed that the acidic residues (HPLDETP) did not belong to any motif.

7. #Line 214, Figure 4: Is it possible to group as stress, hormone, biological etc? Author can add this one in the supplementary file.

Response: Thanks for your suggestion. We have added the Supplementary Figure 2, in which these grouped data are presented along with appropriate explanations (Shown in lines 222-224, and lines726-727)

8. #Lines 330, and so on: Please write “cis-regulatory elements” instead of “cis- regulatory element” throughout the manuscript.

Response: Thanks for your suggestion. We have corrected it.

         We hope you will agree that we have addressed all the concerns raised in your previous review of the manuscript and improved the manuscript. Thank you for your re-consideration of our manuscript and we are looking forward to your favorable decision.